# Oxidative Stress Is a Potential Cost of Synchronous Nesting in Olive Ridley Sea Turtles

**DOI:** 10.3390/antiox11091772

**Published:** 2022-09-08

**Authors:** B. Gabriela Arango, David C. Ensminger, Diana Daniela Moreno-Santillán, Martha Harfush-Meléndez, Elpidio Marcelino López-Reyes, José Alejandro Marmolejo-Valencia, Horacio Merchant-Larios, Daniel E. Crocker, José Pablo Vázquez-Medina

**Affiliations:** 1Department of Integrative Biology, University of California, Berkeley, Berkeley, CA 94720, USA; 2Department of Biological Sciences, San José State University, San Jose, CA 95192, USA; 3Centro Mexicano de la Tortuga, Oaxaca 70947, Mexico; 4Campamento Tortuguero Palmarito, Oaxaca 70984, Mexico; 5Facultad de Medicina Veterinaria y Zootecnia, Universidad Autónoma Benito Júarez de Oaxaca, Oaxaca 68120, Mexico; 6Instituto de Investigaciones Biomédicas, Universidad Nacional Autónoma de México, Ciudad de México 04510, Mexico; 7Department of Biology, Sonoma State University, Rohnert Park, CA 94928, USA

**Keywords:** reproduction, life history theory, metabolic cost, arribada, oxidative damage

## Abstract

Olive ridley sea turtles, *Lepidochelys olivacea*, exhibit a polymorphic reproductive behavior, nesting solitarily or in mass aggregations termed “arribadas”, where thousands of individuals nest synchronously. Arribada nesting provides fitness benefits including mate finding during nearshore aggregations and predator satiation at the time of hatching, but it is unknown if such benefits come with a physiological cost. We used plasma metabolite profiling, stable isotope analysis, biochemical and endocrine assays to test whether metabolic parameters differ between nesting modes, and if arribada nesting is associated with increased levels of oxidative damage compared to solitary nesting. Arribada nesters were bigger and had higher circulating thyroid hormone levels than solitary nesters. Similarly, pathways related to phospholipid and amino acid metabolism, catabolic processes, and antioxidant defense were enriched in individuals nesting in arribada. Stable isotope signatures in skin samples showed differences in feeding zones with arribada nesters likely feeding on benthic and potentially more productive grounds. Arribada nesters had increased levels of plasma lipid peroxidation and protein oxidation products compared to solitary nesters. These results suggest that metabolic profiles differ between nesting modes and that oxidative stress is a trade-off for the fitness benefits associated with arribada nesting.

## 1. Introduction

Life history theory posits that reproductive effort negatively affects survival, but there is conflicting evidence about the proximal costs of reproduction [1]. One of the leading hypotheses suggests that oxidative stress is a trade-off for reproductive investment [2,3,4], but other studies do not support this idea [5,6]. Work in wild vertebrates shows that diversification of reproductive strategies is associated with sexual differences in oxidative stress [7,8,9]. In male free-ranging macaques [10], male northern elephant seals [11], and female North American red squirrels [12], reproductive activities increase oxidative damage. Therefore, the role of oxidative stress as a proximal cost of reproduction is likely species- and sex-specific and also varies among different reproductive strategies.

Sea turtles of the genus *Lepidochelys* spp. exhibit a polymorphic reproductive behavior, nesting both solitarily and in massive aggregations termed “arribadas” [13]. During arribada nesting, thousands of individuals nest synchronously over a two to seven day period [13,14,15]. Notably, the same individual can nest interchangeably in arribadas or solitarily; however, the factors that determine whether an individual joins the arribada or nests in solitary remain unknown [14,16].

Arribada nesting provides a social context that promotes multiple mating and paternity, increasing genetic exchange by three-fold compared to solitary nesting [14,17,18,19]. Arribada nesting also increases early hatchling dispersal by overwhelming or satiating predators [20]. Arribada nesting, however, does not promote hatchling success as increased organic matter from egg saturation and high nesting density in arribada beaches decrease hatchling success compared to solitary nesting [15,21,22]. Similarly, arribada nesters produce smaller hatchlings with lower male-to-female ratio than solitary nesters [13,23]. Thus, while both nesting modes are likely important for the population, it is unknown if the fitness benefits associated with arribada nesting come with a physiological cost.

Here, we compared circulating concentrations of reproductive and metabolic hormones, primary metabolites, stable isotope signatures, and markers of oxidative stress between olive ridley sea turtles (*Lepidochelys olivacea*) nesting solitarily and in arribada. We found that arribada nesters are bigger and likely have higher metabolic activity than solitary nesters. We also found higher circulating levels of lipid peroxidation and protein oxidation products in arribada than in solitary nesters. Our results suggest that oxidative stress is a potential cost for the fitness benefits associated with arribada nesting in olive ridley sea turtles.

## 2. Materials and Methods

### 2.1. Animals and Sample Collection

Animal handling protocols were approved by Sonoma State University. Samples were collected under permit SGPA/DGVS/12915/16 and imported to the US under permits CITES MX88143 and CITES 19US85728C/9. Nesting olive ridley sea turtles were sampled at the marine protected area of *La Escobilla*, Oaxaca, Mexico (15°47′ N; 96°44′ W) throughout the arribada (*n* = 13). Solitarily nesting individuals were sampled at *Campamento Tortuguero Palmarito*, Puerto Escondido, Oaxaca, México (15°53′26.3″ N; 97°07′52.2″ W), and *La Escobilla* (*n* = 10). Samples were collected after the animals had dug their nest, during the ‘trance-nesting period’ [24]. None of the animals were disturbed from nesting, nor returned to the ocean without laying their eggs. Mass was estimated using a regression from published olive ridley morphometric data (*n* = 59, mass = −47.44 + 1.13 × straight carapace length (SCL), r2 = 0.70, *p* < 0.001; [25]). Animals sampled during solitary nesting were also weighed using a hand-held scale (±0.1 kg) to assess the validity of the mass-estimation method. The equation used to estimate mass predicted the weight of solitary nesters with a mean error of 4%. ~25 mL of blood were collected from the cervical vein into chilled Vacutainer tubes [24,26]. Plasma and serum were prepared by centrifugation onsite, frozen, and subsequently stored at −20 °C until laboratory analysis. Epidermal tissue samples were collected using a 6.0 mm diameter biopsy punch (Miltex, York, PA, USA) and stored at −20 °C.

### 2.2. Biochemical Assays

Hormones: progesterone (P4), estradiol (E2), thyroxine (T4), triiodothyronine (T3) and testosterone were measured in serum using commercially available kits (P4: MP Biomedical ELISA catalog number 07BC1113, E2: MP Biomedical ELISA catalog number 07BC1111, T3: MP Biomedical RIA catalog number 06B-254215, T4: MP Biomedical RIA catalog number 06B-254011 (MP Biomedical, Irvine, CA, USA), testosterone: Enzo ELISA catalog number ADI-900-065 (Enzo, Farmingdale, NY, USA)). The assays were validated for use in olive ridley sea turtles. Serially diluted pooled samples (1:2 to 1:16) exhibited parallelism to the standard curve after log-logit transformation. Mean recovery of hormone added to serum pools was 101.5 ± 3.5%, 98.3 ± 6.5%, 99.4 ± 4.7%, and 101.3 ± 5.1% (r2 > 0.98) for P4, E2, T3 and T4, respectively. The testosterone kit was previously validated for use in sea turtle blood [27] and was further validated in our samples via parallelism and spike recovery. Assays were conducted following the manufacturers’ instructions. The glucose, lactate, and corticosterone concentrations included in our correlation analyses were measured in samples obtained from the same animals in our previous study [28].

Oxidative damage: Two markers of lipid peroxidation (4-hydroxynonenal: 4-HNE, malondialdehyde: MDA), a marker of protein oxidation (protein carbonyls), and a marker of protein nitration (3-nitrotyrosine) were measured in plasma using ELISA kits (Cell Biolabs catalog numbers: STA-838, STA-832, STA-310, STA-305, Cell Biolabs, San Diego, CA, USA). Assays were conducted following the manufacturer’s instructions with minor modifications as described in our previous work [29].

Plasma lipids and total protein: Plasma triglycerides (TG) and non-esterified fatty acids (NEFA) were measured using colorimetric kits (TG: Cayman Chemical catalog number 10010303 (Cayman Chemical, Ann Arbor, MI, USA), NEFA: Wako Chemicals, HR Series NEFA-HR (2) catalog number: 999-34691 (Wako Chemicals, Richmond, VA, USA)). Total protein content was measured using a Rapid Gold BCA protein assay kit (Thermo Fisher Scientific catalog number: A53227 (Thermo Fisher Scientific, Waltham, MA, USA)). Oxidative damage values were normalized to total plasma protein levels.

All samples were analyzed in duplicate in a single assay. The average intra-assay coefficient of variation was <6%.

### 2.3. Metabolite Profiling

Analysis of primary metabolism by ALEX-CIS gas chromatography-mass spectrometry (GC-TOF MS) was conducted in plasma at the UC Davis West Coast Metabolomics Center following the methods of Fiehn et al. [30]. Raw data were pre-processed and stored as apex masses, exported to a data server with absolute spectra intensities, and further filtered with an algorithm implemented in the BinBased database [30]. Spectra were cut to a 5% base peak abundance and matched to a database entry. Quantification was stored as peak height using the unique ion as default for all database entries that are positively detected in more than 10% of the unidentified metabolites.

### 2.4. Stable Isotopes

Epidermal tissue samples were rinsed with DI water and dried at 60 °C for 48 h. Samples were grounded, weighted to ~1mg, and packed into tin capsules (3.5 × 5 mm, #041060, Costech Analytical Technologies, Valencia, CA, USA). Samples were analyzed for %C, δ^13^C, %N, and δ^15^N by continuous flow dual isotope analysis using a CHNOS Elemental Analyzer interfaced to an IsoPrime100 mass spectrometer at the UC Berkeley Center for Stable Isotope Biogeochemistry. Stable isotope ratios are expressed in δ notation as parts per thousand (‰). Long-term external precision for C and N isotope determinations was ±0.10‰ and ± 0.20‰, respectively. C:N ratios in both groups were <3.5, validating δ^13^C values by indicating a low lipid content.

### 2.5. Statistical Analyses

Biochemical assays: statistical analyses were conducted using JMP Pro 15 (SAS Institute, Cary, NC, USA). Equality of variances was assessed using Levene’s test. P4, TG and NEFA data were log10 transformed to meet model assumptions. Arribada and solitary nesting groups were compared using two-sample *t*-tests. A nominal logistic regression was used to analyze 4-HNE data since the response variable was binary. Correlation analyses between hormones, mass, TG, NEFA, glucose and lactate were conducted by calculating Spearman rank correlation coefficients. The glucose, lactate, and corticosterone concentrations included in our correlation analyses were measured in samples obtained from the same animals in our previous study [28]. Statistical significance was considered at *p* ≤ 0.05.

Metabolite profiling: statistical analysis was conducted using MetaboAnalyst 5.0 (Xia et al.; Main server: https://www.metaboanalyst.ca, 6 September 2022). Metabolite peaks were normalized using the median of all peak heights for all identified metabolites. To minimize noise, low quality peaks (0–5% of the mean) were filtered out [31]. Normalized peaks were log10 transformed to meet model assumptions. Data were compared between arribada and solitary using multiple *t*-tests with a 5% FDR adjustment. A heatmap was constructed using Pearson distance correlation [32] and the average clustering method for the top 50 most abundant metabolites. Quantitative enrichment analysis was conducted in MetaboAnalyst using the small molecule pathway database (SMPDB) [33].

Stable isotopes: Variables were compared between nesting modes using two-sample *t*-tests. Ratios of δ^13^C and δ^15^N were plotted against each other with standard ellipses with 40% of the data using ggplot2 in R.

## 3. Results

### 3.1. Biochemical Profiles

We compared circulating TG, NEFA, P4, E2, testosterone, T3 and T4 concentrations between solitary and arribada nesters to determine whether hormonal and biochemical profiles differ between nesting modes. While neither TG nor NEFA were different between arribada and solitary nesters (TG: solitary 14.46 ± 9.52 vs. arribada 20.47 ± 10.03 mg/mL, t = 1.94, df = 21, *n* = 23, *p* = 0.07; NEFA: solitary 0.93 ± 0.22 vs. arribada 1.08 ± 0.55 mM, t = 0.445, df = 22, *n* = 24, *p* = 0.66, Appendix A Figure A1a,b), P4 was higher in arribada than in solitary nesters (3.58 ± 1.71 vs. 2.10 ± 1.56 ng/mL, t = 2.612, df = 21, *n* = 23, *p* = 0.02; Figure 1a). E2 did not differ between nesting conditions (solitary 4.47 ± 2.79 vs. arribada 5.7 ± 2.24 pg/mL, t = −1.17, df = 21, *n* = 23, *p* = 0.25; Figure 1b). Similarly, testosterone was not different between solitary and arribada nesters (70.43 ± 42.88 vs. 75.08 ± 35.99 pg/mL, t = −0.29, df = 22, *n* = 24, *p* = 0.78; Figure 1c). Both T3 and T4 were higher in individuals nesting in arribada than in solitary nesters (T3: 48 ± 19.54 vs. 29.3 ± 11.86 ng/dL, t = 2.66, df = 21, *n* = 23, *p* = 0.015; T4: 1.45 ± 0.27 vs. 1.12 ± 0.13 ug/dL, t = 3.62, df = 21, *n* = 23, *p* = 0.002; Figure 1d,e). Mass was also higher in arribada than solitary nesters (29.20 ± 5.20 vs. 24.76 ± 4.37 kg, t = 2.17, df = 21, *n* = 23, *p* = 0.042, Appendix A Figure A1c). These results show that arribada nesters are heavier and have higher T3, T4 and P4 but not E2 or testosterone levels than solitary nesters.

We then conducted correlation analyses using mass, hormones, TG, NEFA, glucose, and lactate [28]. Glucose and lactate showed a positive association in both nesting modes; however, this association was stronger in arribada (r_s_ = 0.82, *p* = 0.0006) and not significant in solitary nesters (r_s_ = 0.60, *p* = 0.067) (Figure 2a,b). The strongest positive associations were observed between corticosterone and glucose (r_s_ = 0.87, *p* < 0.001, corticosterone and lactate (rs = 0.73, *p* = 0.004), and TG and NEFA (r_s_ = 0.75, *p* = 0.003) in arribada nesters (Figure 2b). These associations were not observed in solitary nesters, which showed positive associations between progesterone and estradiol (r_s_ = 0.71, *p* = 0.020), TG and mass (r_s_ = 0.77, *p* = 0.009), TG and T4 (r_s_ = 0.64, *p* = 0.048), and a negative correlation between TG and lactate (r_s_ = −0.66, *p* = 0.037) (Figure 2a). These results show that arribada and solitary nesters have different blood biochemical profiles. Furthermore, these results suggest that metabolic activity might be higher in arribada than in solitary nesters.

### 3.2. Metabolite Profiling

We conducted an analysis of primary metabolism by GC-TOF MS to further explore potential differences in circulating metabolites between nesting modes. Our analysis detected 481 metabolites including 155 known metabolites which clustered based on nesting mode (Figure 3a), further suggesting that metabolism differs between solitary and arribada nesters. Of the identified known metabolites, 10 were significantly expressed (FDR 5%) between nesting modes, with five downregulated and five upregulated in arribada compared to solitary nesters (Figure 3b). The top downregulated metabolite in arribada nesters was phosphoethanolamine, while the top upregulated metabolite was glutamic acid. Accordingly, enrichment analysis showed over-representation of pathways related to phospholipid (phosphatidylcholine and phosphatidylethanolamine biosyntehesis, sphingolipid metabolism), and amino acid metabolism (beta-alanine, tryptophan, tyrosine, and glutamate metabolism), catabolic processes including malate-aspartate shuttle and the glucose-alanine cycle, and glutathione (GSH) metabolism. Of note, two of the most downregulated metabolites in arribada compared to solitary nesters were the antioxidants uric acid and α-tocopherol. These results further suggest that arribada nesters have increased metabolic activity compared to solitary nesters. Moreover, these results suggest that there are differences in redox metabolism between nesting modes.

### 3.3. Stable Isotopes

We conducted stable isotope analysis to detect potential differences in resource acquisition between nesting modes. We found significant differences in %C (t = 2.74, df = 18, *n* = 20, *p* = 0.013) and %N (t = 2.73, df = 18, *n* = 20, *p* = 0.014) between nesting modes, but not in C:N ratios (Figure 4a). When comparing stable isotope signatures, we also found small overlaps between nesting modes on 40% ellipses. Our results suggest that before coming to nest, individuals nesting in arribada are likely feeding at lower trophic levels and in more productive benthic feeding grounds than individuals nesting solitarily (Figure 4b).

### 3.4. Oxidative Damage

We evaluated whether oxidative damage represents a potential proximate cost for nesting in arribada by comparing four circulating markers of oxidative damage among nesting modes. Plasma levels of the lipid peroxidation products 4-HNE (arribada 100% positive vs. solitary 25% positive for 4-HNE, χ^2^(2) = 10.01, *n* = 17, *p* = 0.0016), and MDA (0.0025 ± 0.0013 vs. 0.0011 ± 0.00034 pmol/μg of protein, t = 3.77, df = 15, *n* = 17, *p* = 0.0019), along with protein carbonyls (0.00246 ± 0.0013 vs. 0.00103 ± 0.00032 nmol/μg of protein, t = 3.64, df = 14, *n* = 16, *p* = 0.0027) were higher in arribada than in solitary nesters (Figure 5a–c). In contrast, protein nitration (3-Nitrotyrosine) did not vary between nesting modes (solitary 0.039 ± 0.026 vs. arribada 0.058 ± 0.031 nM/mg, t = −1.29, df = 15, *n* = 17, *p* = 0.22; Figure 5d). These results suggest that oxidative stress is a potential cost of arribada nesting in olive ridley sea turtles.

## 4. Discussion

Potential fitness benefits associated with arribada nesting include mate finding during nearshore aggregations [14], predator satiation at the time of hatching [20], multiple paternity and increased genetic exchange [19]. Whether this specialized nesting mode carries a physiological cost was previously unknown. In this study, we found that arribada and solitary nesters have distinct circulating metabolic profiles and that arribada nesters are heavier and have higher P4, T3, T4, lipid peroxidation, and protein oxidation levels than solitary nesters. We also found differences in stable isotope signatures between nesting modes and enrichment for catabolic and antioxidant pathways in arribada compared to solitary nesters. Therefore, our results suggest that nesting in arribada might be energetically more expensive than nesting solitarily and that oxidative damage is a potential trade-off for the fitness benefits associated with arribada nesting in olive ridley sea turtles.

Little is known about the endogenous adjustments that allow individuals to synchronize to join an arribada. *Lepidochelys* spp. can retain their eggs for longer periods compared to other species of sea turtles [34]. Similarly, tolerance to hypoxia-induced pre-ovipositional embryonic arrest is higher in eggs from arribada than from solitary nesters [23]. Both increased egg retention capacity and embryonic arrest appear to be important to synchronize to join the arribada. Similarly, gonadosteroids might also influence whether an individual joins the arribada. We found higher P4 but not E2 or testosterone levels in arribada than in solitary nesters. A P4 surge during ovulation induces a rapid albumen release into the oviduct, which activates previously stored sperm [35]. Therefore, it is possible that higher P4 levels in arribada than in solitary nesters are related to increased mating opportunities during the arribadas. Of note, P4 levels do not differ between nesting modes in Kemp’s ridleys [36]. Thus, there is no conclusive evidence about the role of gonadosteroids in promoting arribada nesting in *Lepidochelys* spp.

In a previous study we showed that arribada nesters have higher circulating corticosterone and glucose levels than solitary nesters [28]. Here, we found that arribada nesters are bigger and have higher thyroid hormone levels than solitary nesters. Moreover, we found strong positive correlations between corticosterone and glucose, corticosterone and lactate, glucose and lactate, and TG and NEFA in arribada but not in solitary nesters. Similarly, metabolite profiling shows differences in major pathways related to cell metabolism and antioxidant defense. Phosphoethanolamine (PETh) was the most downregulated metabolite in arribada compared to solitary nesters. Besides being important for cell membrane composition, PETh stimulates tolerance to nutrient starvation, and its levels increase in glutamine-deprived cells [37]. Consistent with this observation, levels of glutamic acid, 4-aminobutyric acid, pantothenic acid, aspartic acid, and glyceric acid were increased in arribada compared to solitary nesters, while levels of antioxidants α-tocopherol, uric acid, picolinic acid and, 5-methoxytryptamine were decreased. These metabolites are important for insulin production, glycolysis, and redox balance [38].

We also found enrichment for catabolic processes, including the glucose-alanine cycle and the malate-aspartate cycle. Of note, both of these pathways produce oxidants as byproducts either directly or through downstream effectors [39,40]. Thus, our results support the idea that arribada nesters have higher metabolic activity than solitary nesters. Similarly, our combined results also suggest that arribada nesters have larger energy reserves than solitary nesters. Our results are consistent with those reported in animals nesting in the Rushikulya Rookery of Orissa, India, where arribada nesters are also larger than solitary nesters [41]. As discussed earlier, the same individual can nest interchangeably in arribadas or in solitary, but the factors that determine whether an individual joins the arribada or nests solitarily remain unknown [14,16]. Our results suggest that arribadas are potentially more energetically costly than solitary nesting and that bigger animals with larger energetic reserves join the arribadas.

As capital breeders [42,43,44], sea turtles feed and build their energy reserves prior to migrating, mating, and nesting. Hence, if arribada nesting is more energetically costly than solitary nesting, individuals lacking appropriate energy reserves might choose solitary nesting over joining the arribadas despite losing the fitness benefits associated with arribada nesting. Solitary nesting requires shorter inter-nesting intervals. Thus, solitary nesting likely results in less time spent away from the feeding grounds [45], potentially allowing solitary nesters to build their energy reserves and join the arribadas during subsequent reproductive bouts. Our stable isotope analysis shows that solitary and arribada nesters likely feed in different grounds before arrival to the nesting beach. Both δ^13^C and δ^15^N values are consistent with reported data for this East Pacific population [46]. Although we were not able to measure prey items, the ellipse distances in δ^13^C and δ^15^N suggest that solitary and arribada nesters feed at different trophic levels and benthic zones. More specifically, the higher δ^15^N values seen in solitary as opposed to arribada nesters suggest that solitary nesters feed at a higher trophic level than arribada nesters [47]. Moreover, the less negative δ^13^C values observed in arribada than in solitary nesters suggest that arribada nesters feed in benthic and more productive feeding grounds than solitary nesters [47]. Therefore, differences in resource availability and allocation might ultimately affect whether an individual joins the arribada. According to life history theory, reproduction is a costly life history trait, and our results suggest that nesting in arribada is more energetically expensive than nesting solitarily, though it carries fitness benefits such as increased genetic exchange [19].

The cost of reproduction represents one of the most fundamental life history trade-offs [3], but there is inconclusive evidence about whether oxidative stress is a proximal cost of reproductive investment [1]. Work with laboratory versus wild animals often yields conflicting results [48]. Similarly, differences in reproductive strategies within wild vertebrates likely result in differential susceptibility to oxidative stress [7,8,9]. Whether capital breeders have higher susceptibility to oxidative stress than income breeders remains understudied. In macaques, North American red squirrels, aspic vipers and northern elephant seals, reproductive activities increase oxidative damage despite concurrent increases in antioxidant defenses [10,11,12,49]. Here we found higher lipid peroxidation and protein oxidation levels in arribada compared to solitary nesters, and reduced levels of α-tocopherol and uric acid, a primary circulating antioxidant which varies with stress levels [50]. We also found that arribada nesting is likely more energetically costly than solitary nesting. These results suggest that higher energy expenditure in olive ridley turtles nesting in arribadas is associated with increased oxidative stress. In northern elephant seals, breeding increases circulating lipid peroxidation in males but not in females [11]. Elephant seals are polygynous, sexually dimorphic capital breeders [51,52]. Males compete for position in a dominance hierarchy used to control access to females [51,52,53,54]. Thus, energy expenditure associated with breeding is higher in male than in female elephant seals [55]. Hence, it is possible that in capital breeders with unique reproductive behaviors that result in increased energy expenditure, such as male elephant seal males or female olive ridleys nesting in arribada, oxidative stress is a trade-off for the fitness benefits associated with such behaviors.

## 5. Conclusions

We found that arribada nesting in olive ridley sea turtles is associated with increased levels of circulating oxidative damage and reduced antioxidant levels compared to solitary nesting. We also found that biochemical, endocrine and metabolomic profiles and stable isotope signatures differ between nesting modes, with arribada nesters likely having increased energy reserves and metabolic activity compared to solitary nesters. These results suggest that oxidative damage is a potential cost of synchronous nesting in olive ridley sea turtles. As such, oxidative stress may be a trade-off for a reproductive mode that carries increased fitness benefits in a capital breeder. Whether these trade-offs are present in other capital breeders remains unknown and warrants further investigation.

## Figures and Tables

**Figure 1 antioxidants-11-01772-f001:**
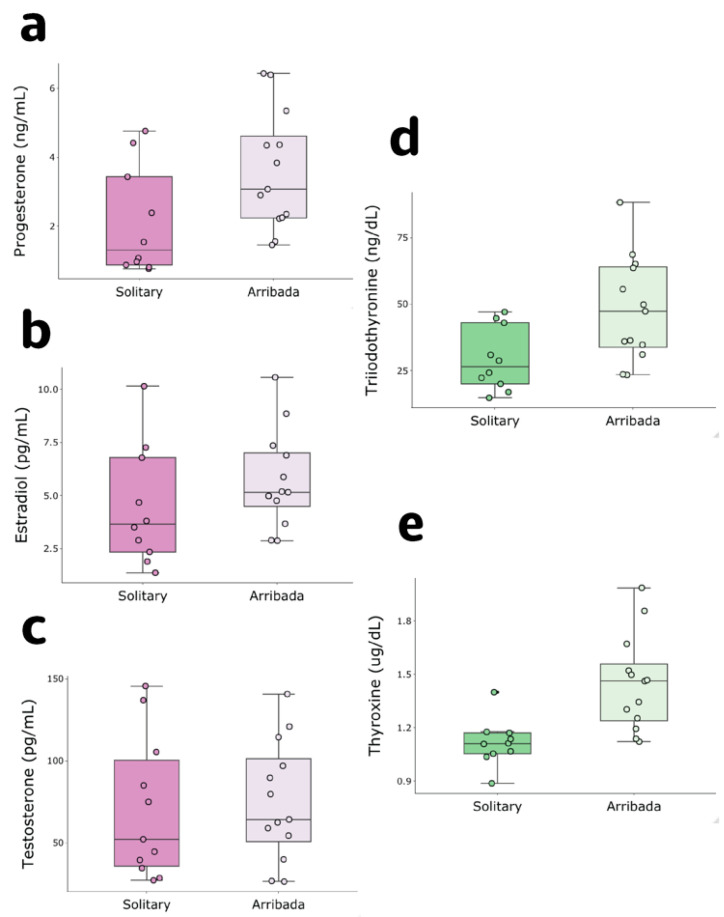
Reproductive and metabolic hormone levels in solitary and arribada nesters. (**a**) Progesterone (*p* = 0.02), (**b**) estradiol (*p* = 0.25), (**c**) testosterone (*p* = 0.78), (**d**) triiodothyronine (*p* = 0.015), and (**e**) thyroxine (*p* = 0.002).

**Figure 2 antioxidants-11-01772-f002:**
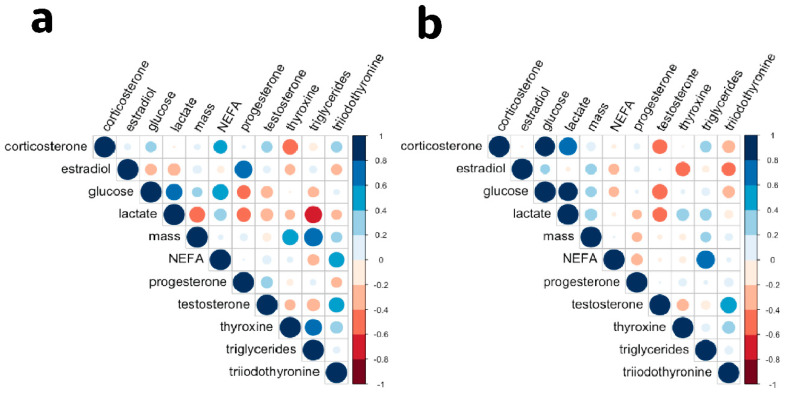
Correlation analysis between hormones and metabolites in (**a**) solitary and (**b**) arribada nesters.

**Figure 3 antioxidants-11-01772-f003:**
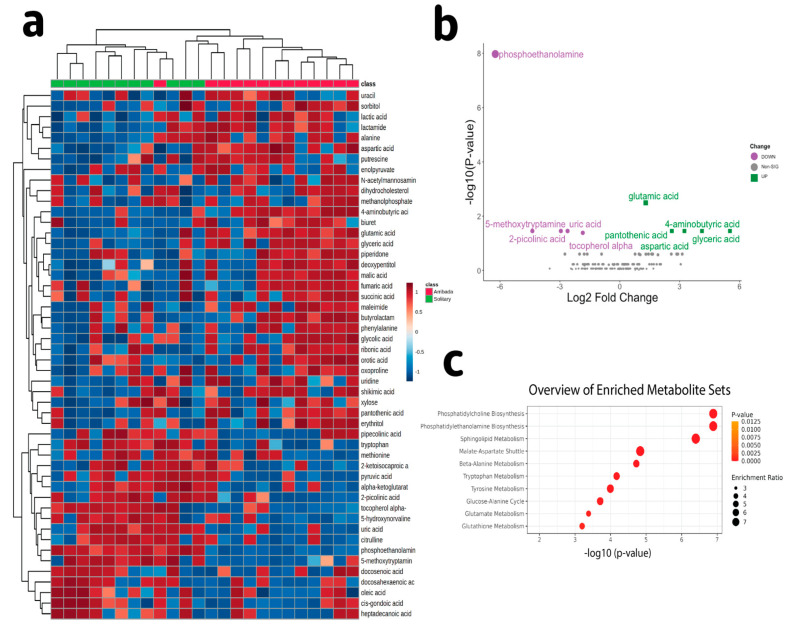
Metabolite profiling. (**a**) Heat map showing the 50 most abundant metabolites in olive ridley plasma, (**b**) Volcano plot showing differentially expressed metabolites in arribada compared to solitary nesters (5% FDR), and (**c**) Enrichment pathways showing significantly enriched pathways in arribada compared to solitary nesters.

**Figure 4 antioxidants-11-01772-f004:**
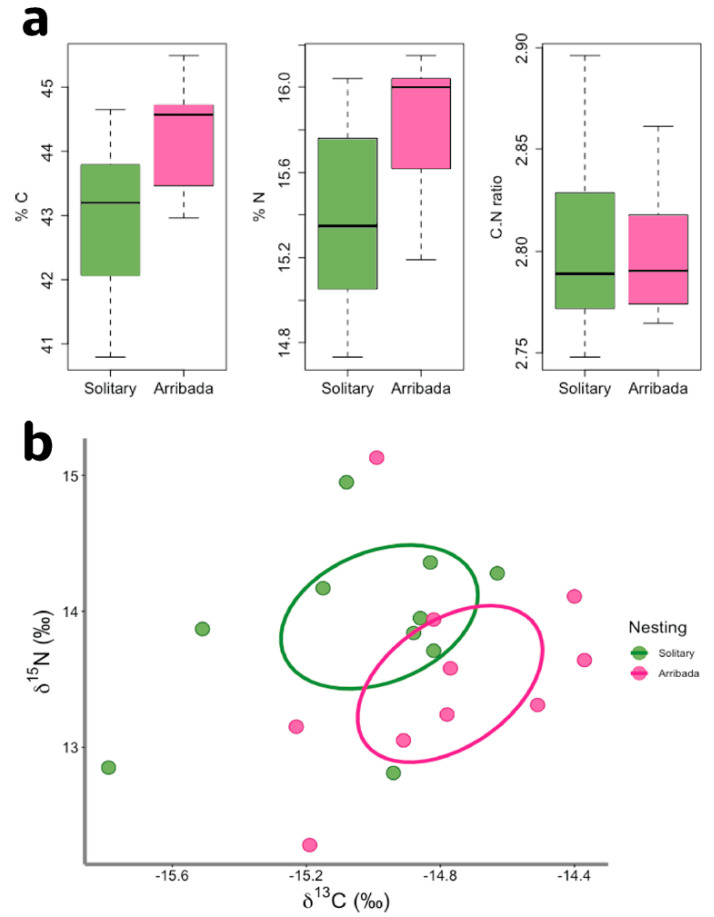
Stable isotope analysis. (**a**) Comparison of %C (*p* = 0.013) and %N (*p* = 0.014) and C:N (*p* > 0.05) ratio between solitary and arribada nesters, (**b**) δ^13^C and δ^15^N ratios. Standard ellipses represent a 40% overlap between nesting modes.

**Figure 5 antioxidants-11-01772-f005:**
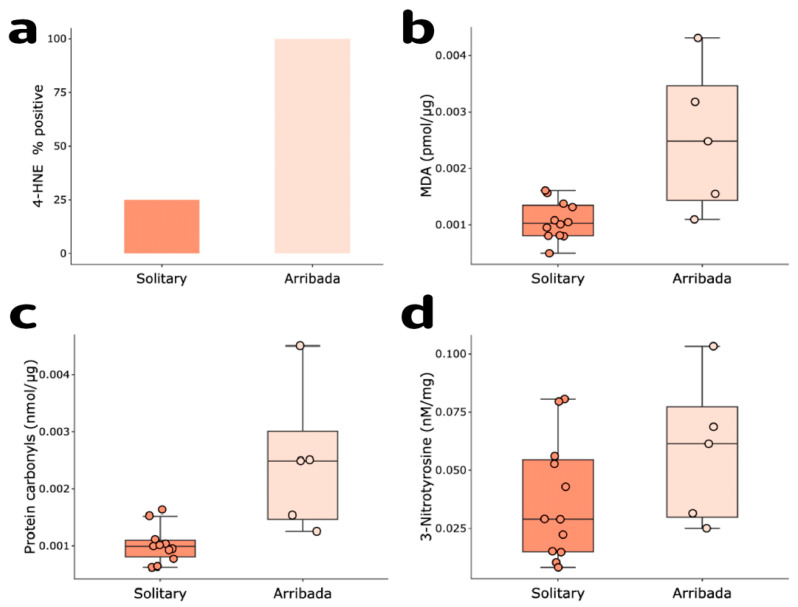
Oxidative damage. Lipid peroxidation products: (**a**) 4-HNE-protein adducts (binary test, percent positive arribada, 100% vs. solitary, 25%, *p* = 0.0009, χ^2^(2) = 10.01, *p* = 0.0016), and (**b**) MDA-protein adducts (*p* = 0.0019); (**c**) protein carbonyls (*p* = 0.0027) and (**d**) 3-Nitrotyrosine (*p* = 0.22).

## Data Availability

The datasets generated for this study can be found in the figshare https://figshare.com/articles/dataset/Arango_etal_OR_DATA_xlsx/14864427 (accessed on 6 September 2022).

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
