# Peer review of "Oxidative Stress Is a Potential Cost of Synchronous Nesting in Olive Ridley Sea Turtles"

_antioxidants, 2022, doi:10.3390/antiox11091772_

Round 1

Reviewer 1 Report

The article is very interesting and novel in this field but I find it too descriptive for this journal. Furthermore, only a few biomarkers of oxidative damage are determined without focusing on the antioxidant mechanism by which these biomarkers decrease. Only plasma GSH levels are named but not measured in both groups as the main antioxidant.

Figure 5.A. requires the error bars and sample points.

Reviewer 2 Report

In this manuscript, Arango et al. describe the metabolite, stable isotope, biochemical, and endocrinal profiles of olive ridley sea turtles nesting as an individual or in arribadas. The authors identified that individuals in arribadas are heavier and more fit however they present higher oxidative stress. This work is interesting and I support publication in Antioxidants, once all the following comments are addressed.

  • In Figure 3, the “a, b, and c” labels should be moved to the left of their corresponding panel.
  • Why were the glucose, lactate, and corticosterone concentration used in the correlation analysis measured in samples from a previous study by the same authors? Why were the authors unable to measure these metabolites in this new study? I am not convinced that these data can simply be taken from a different study.
  • All the data presented in section 3.1 should be plotted and shown either in Figure 1 or in a supporting figure.
  • The authors report an increase in oxidative stress that was measured via ELISA assays. Do the authors observe changes in metabolites linked to oxidative stress measured via their metabolite profiling and dataset analysis? Why do the authors not see changes in oxidative stress/antioxidant response in their pathway analysis? This is very confusing.

Reviewer 3 Report

This is well written paper.

MAJOR CONCERNS

None

MINOR ISSUES

1) The font used for illustrations is very small.

2)  If individual panels of figures are identified as a, b, c, d, etc., the corresponding legends should also refer to a, b, c, d (lower case) rather than A, B, C, D (capitalized).

3) Fig. 3C (b):  Symbols for Non-SG (faint grey) are almost invisible.

4) Some Greek letters are in bold for no apparent reason (see for example lines 229, 230, 231.

5) Ref. 7: "Fish Fish" should probably be changed to "Fish and Fisheries".

Round 2

Reviewer 1 Report

I consider that the work has been improved and is justified.

Reviewer 2 Report

The authors addressed all of my concerns with this revised manuscript. I now recommend publication in antioxidants.